# Barriers and Enablers to Health-Seeking for People Affected by Severe Stigmatising Skin Diseases (SSSDs): A Scoping Review

**Rosalind McCollum [1,*], Hannah Berrian [2], Sally Theobald [1], Zeela Zaizay [3], Karsor Kollie [4] and Laura Dean [1]**

[1] Department for International Public Health, Liverpool School of Tropical Medicine, Liverpool L3 5QA, UK; sally.theobald@lstmed.ac.uk (S.T.); laura.dean@lstmed.ac.uk (L.D.)

[2] Institute for Research and Evaluation, University of Liberia-Pacific, Monrovia 1000, Liberia; hannahberrian@gmail.com

[3] Actions Transforming Lives, Monrovia 1000, Liberia; zeelazaizay@gmail.com

[4] Neglected Tropical Disease Programme, Ministry of Health Liberia, Monrovia 1000, Liberia; karsorkollie@gmail.com

\* Correspondence: rosalind.mccollum@lstmed.ac.uk

**Abstract:** People affected by severe stigmatising skin diseases (SSSDs) often live in the poorest communities, within the poorest countries, and experience a range of barriers to seeking timely, quality care. This scoping review analyses the available literature on health-seeking for patients affected by SSSDs, to identify enablers and barriers to health-seeking. We searched MEDLINE complete, CINAHL, Global Health databases for suitable articles published between 2010 and 2020. Search strings were compiled for health-seeking, SSSDs and lower middle-income countries (LMIC). Our search returned 1004 studies from across three databases. Of these, 136 potentially relevant studies were identified and full texts were reviewed for eligibility against the inclusion criteria, leading to the inclusion of 55 studies. Thematic narrative analysis was used, with results framed around the Levesque framework to analyse barriers and enablers to health-seeking along the continuum of the patient pathway. This scoping review has revealed barriers across the patient pathway, from both supply and demand aspects of health services. Spiritual beliefs emerged strongly relating to care-seeking and underlying stigma. Curative care was a focus for the majority of studies, but few papers emphasised holistic care (such as physical rehabilitation and psychosocial support). From our analysis, greater community engagement is needed to reduce barriers along the patient-care pathway.

**Keywords:** health-seeking; neglected tropical diseases; stigma; skin disease; holistic care

## 1. Introduction

Neglected tropical diseases (NTDs) are all preventable, and to varying degrees treatable (Sun and Amon 2018). NTDs form key tracers for identifying disparities in progress towards universal health coverage (UHC) and equitable access to high-quality health services (WHO 2020). Despite being preventable and treatable, NTDs still affect around one billion people worldwide, predominantly those who are most disadvantaged and marginalised, living in the poorest countries and within the poorest communities in these countries (Sun and Amon 2018; Kuper 2019). We already know that people who are the most marginalised tend to experience the greatest health inequities (WHO 2008), with experience of barriers to seeking and using available health services greatest among those who are most disadvantaged.

NTDs are often reported, diagnosed and treated late, by which time they have often led to chronic conditions and disability (Kuper 2019). It is therefore critical to identify the barriers along the patient pathway in order to promote earlier perception of need, health care-seeking, reaching, payment and engagement with available services, and health systems' ability to deliver services which are approachable, acceptable, available and

accommodating of patient needs, affordable and appropriate for each patient (Levesque et al. 2013). For patients with NTDs, this includes accurate diagnosis and quality treatment before disability occurs (where possible), ongoing care related to any disability (where needed), psychological support and interventions aimed at reducing stigma as part of holistic care for people affected (WHO 2020).

## 2. Rationale for the Study

This study forms part of the REDRESS research consortium. REDRESS aims to help improve the care of people affected by severe stigmatising skin diseases (SSSDs), a subgroup of NTDs, in Liberia. SSSDs in this study refer to leprosy, onchocerciasis, Buruli Ulcer, yaws, and clinical presentation of lymphatic filariasis, specifically lymphoedema and hydrocele. For many persons with SSSDs, lack of timely access to quality health services results in significant physical and psycho-social consequences, complex treatment journeys, and large economic impacts. We therefore sought to conduct a review of the available literature on health-seeking for patients affected by the following SSSDs—leprosy, onchocerciasis, Buruli Ulcer, yaws, and clinical presentation of lymphatic filariasis, specifically lymphoedema and hydrocele, in order to identify enablers and barriers to health-seeking in relation to SSSDs.

## 3. Materials and Methods

This rigorous scoping review aligns with 'Scoping Studies: towards a methodological framework' conducting a scoping study (Arksey and O'Malley 2005), by seeking to map out the relevant literature related to health-seeking for patients affected by SSSDs. We followed the five main stages identified for conducting a scoping study. (1) Identifying the research question; (2) Identifying relevant studies; (3) Study Selection; (4) Charting the data; (5) Collating, summarising and reporting the results (Arksey and O'Malley 2005).

### 3.1. Stage 1: Identifying the Research Question

The research question for this paper is as follows: 'What are the barriers and enablers to health-seeking in relation to NTDs?'. Please see the introduction and rationale sections of this paper for how this question was identified.

### 3.2. Stage 2: Identifying Relevant Studies

We searched MEDLINE complete, CINAHL, Global Health databases for suitable articles published in the past decade (2010–2020). To ensure comprehensive search terms across different aspects of health-seeking, we compiled search strings for the terms below (see Table 1).

**Table 1.** Scoping review search terms.

| Health-Seeking | SSSDs | LMIC |
|:---:|:---:|:---:|
| Access | | |
| Availability | | |
| Help | | |
| Utilisation | Skin disease | |
| Behaviour | Neglected tropical diseases | Sub-Saharan Africa |
| Seek | Leprosy | Africa |
| Practice | Buruli Ulcer | Low and middle income |
| Perception | Yaws | countries |
| Attitude | Lymphoedema | Liberia |
| Belief | Lymphatic filiariasis | Developing countries |
| Enabler | Hydrocele | |
| Barrier | | |
| Demand | | |

Medical subject headings (MeSH) and keywords were used with appropriate Boolean operators (see Appendix A for search terms used). Suitable key words were used in the absence of MeSH terms. The search strategy was developed following initial scoping searches, with search terms refined as needed to ensure that relevant studies are obtained. Finalisation of the search terms was carried out in collaboration with the Liverpool School of Tropical Medicine Library staff. The strategy was initially developed for use with MED-LINE and then modified for use in other databases. All searches were run on 14 May 2020. We also hand searched the reference list of papers that included extensive information on health-seeking and SSSDs for any further relevant papers.

*3.3. Stage 3: Study Selection*

Eligibility criteria applied to select studies for inclusion are described below:

- Any qualitative, quantitative or mixed methods study or review study. No restrictions were placed on study design.
- Study that includes data about barriers/enablers to accessing care for a SSSD. SSSDs included in this study are leprosy, onchocerciasis, Buruli ulcer, yaws, and clinical presentation of lymphatic filariasis, specifically lymphoedema and hydrocele.
- Studies conducted in Lower Middle-Income Countries (LMIC).
- Study written in English language.
- Studies published between 2010 and 2020.

We excluded studies that were opinion pieces or commentaries, to focus on findings from original research and prior reviews. We excluded studies not about barriers/enablers to accessing SSSD care, studies not conducted in LMIC, studies that were published >10 years ago and studies not available in English language. We did not include studies about other potential SSSDs, such as cutaneous leishmaniasis, post-kala azar dermal leishmaniasis, mycetoma, scabies and fungal diseases, due to the need for this literature review to provide knowledge and insights into health-seeking for leprosy, onchocerciasis, Buruli Ulcer, yaws, and clinical presentation of lymphatic filariasis, specifically lymphoedema and hydrocele as part of the wider REDRESS study, which seeks to improve care for people affected by these conditions (which form the central component of care within integrated case management within Liberia).

The search terms used did not include individual country names, with the exception of Liberia, as this is the focal country for REDRESS. This is a limitation of the study and there is the potential that some papers may not have been identified using the broader lower middle-income countries (LMIC) terms used as a result. The search timeframe of 2010 to 2020 was identified for feasibility reasons, in order to summarise the most recent literature available relating to the barriers and enablers to health-seeking, so that the findings from this review could be used to inform the design of interventions, which will be introduced to strengthen holistic care for people affected by these SSSDs during later phases of REDRESS. This is a developing area of research, and as a result findings are continually evolving, and there is an increasing number of studies focusing on this topic. It is a recommendation that the review findings continue to be updated over time.

The search yielded 1004 results across the three databases. These were then imported to Endnote X8 for removal of duplicates, with 128 duplicates removed. Three reviewers (HB, LD, RM) then screened titles and abstracts for relevance, which was tracked through an excel database. In total, 136 full-text papers were sought for studies which were considered potentially relevant, with each study reviewed against the inclusion criteria for inclusion within the review. Titles, abstracts and papers were divided between the three reviewers. Where there was uncertainty about the inclusion of a particular title or abstract, it was included for the next level of review. Where there was uncertainty about the papers reviewed, these were discussed together between all three authors to reach consensus. In total, 55 studies were selected for inclusion (including four identified through reference

list review of suitable papers) (see Figure 1 PRISM chart). All full-text papers reviewed were tracked in an Excel file to document the paper against inclusion criteria.

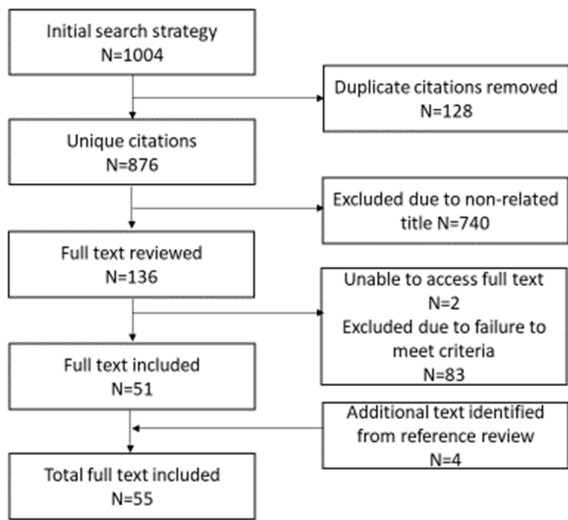

**Figure 1.** PRISM chart.

### 3.4. Stage 4: Charting the Data

Once a suitable paper was identified which met all inclusion criteria, data were extracted. An Excel template was developed based on the Levesque, Harris and Russel, 2013 conceptual framework of access to health care and informed by preliminary reading of data (see Figure 2). We chose to apply this framework for two main reasons: first, the demonstrated applicability of this framework in previous related review studies (Hameed et al. 2020); second, the framework provides a clear distinction of the different potential opportunities for intervention along the patient pathway.

Data quality: Due to the scoping nature of this review, an analysis of quality of the studies was not carried out, as the review sought to identify all suitable studies, not just those considered to be high quality.

Analytical Framework: In order to systematically consider aspects of access, we used the Levesque et al. framework, which views access as "*as the opportunity to identify healthcare needs, to seek healthcare services, to reach, to obtain or use health care services, and to actually have a need for services fulfilled*". We assessed included studies against five demand-side access dimensions (Ability to perceive; Ability to seek; Ability to reach; Ability to pay; and Ability to engage) and five supply-side access dimensions (Approachability; Acceptability; Availability and accommodation; Affordability; Appropriateness) (Levesque et al. 2013).

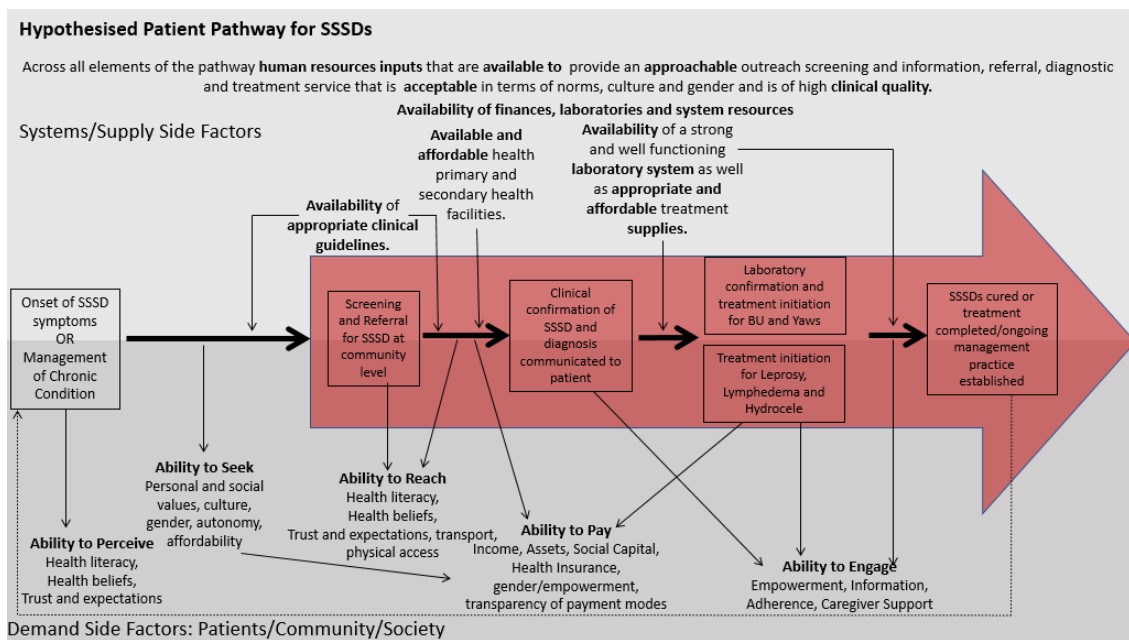

**Figure 2.** REDRESS hypothesised patient pathway for SSSDs. Reprinted/adapted with permission from (Levesque et al. 2013). Patient-Centred Access to Health Care: Conceptualising Access at the Interface of Health Systems and Populations. *International Journal for Equity in Health* 12: 18. https://doi.org/10.1186/1475-9276-12-18. © 2022 Levesque et al.; licensee BioMed Central Ltd.

### 3.5. Stage 5: Collating, Summarising and Reporting the Results

Analysis of the data involved a qualitative thematic narrative analysis, through application of themes identified from the modified framework above, along with an iterative approach to identify and synthesise concepts found in the data. Narratives were developed and data charted based upon the themes arising within the data.

Ethics: No ethical approval was required for this study, since this is a review of existing published studies.

## 4. Results

### 4.1. Study Characteristics

Of the 55 studies included, 30 applied quantitative design, two were review studies, 13 applied a qualitative study design, while eight used mixed methods, one applied modelling methods and one was a report. Table 2 provides an overview of the included papers. The majority of studies (27) included countries from Sub-Saharan Africa; 20 studies focused on countries from Asia (including Western Asia); seven were from the Americas; one had a global focus.

There was wide variation regarding the condition for which health-seeking was studied. By far the majority of studies focused on health-seeking practices for patients with leprosy (33); eight studies focused on care of patients with lymphatic filiariasis/lymphoedema; nine studies for Buruli Ulcer; three studies were for a combination of NTDs; one focused on onchocerciasis; and one for yaws (see Table 2 for details).

Include Table 2 here.

**Table 2.** Overview of included studies.

| Study | Country of Focus | Geographic Region | Chronic Condition/Disease of Focus | Methodology |
|---|---|---|---|---|
| (Abdulmalik et al. 2018) | Nigeria | SSA | Lymphatic Filariasis | qualitative |
| (Abedi et al. 2013) | Iran | Western Asia | Leprosy | qualitative |
| (Ackumey et al. 2011) | Ghana | SSA | Buruli ulcer | quantitative |
| (Adhikari et al. 2015) | Nepal | Asia | Lymphoedema | qualitative |
| (Alferink et al. 2013) | Benin | SSA | Buruli ulcer | quantitative |
| (John et al. 2010) | India | Asia | Leprosy | quantitative |
| (Atre et al. 2011) | India | Asia | Leprosy | mixed |
| (Dean et al. 2019) | Liberia | SSA | Mixed NTDs-lymphatic filariasis, leprosy, Buruli Ulcer, and onchocerciasis | qualitative |
| (Dupnik et al. 2013) | Brazil | South America | leprosy | quantitative |
| (Garchitorena et al. 2015) | SSA | SSA | Buruli ulcer | modelling |
| (Gautham et al. 2011) | India | Asia | Leprosy | quantitative |
| (Girma et al. 2018) | Ethiopia | SSA | leprosy | quantitative |
| (Gómez et al. 2018) | Colombia | South America | leprosy | quantitative |
| (Henry et al. 2016) | Brazil | South America | leprosy | quantitative |
| (Heukelbach et al. 2011) | Brazil | South America | leprosy | quantitative |
| (Ib et al. 2015) | Nigeria | SSA | onchocerciasis | quantitative |
| (Kasang et al. 2019) | Liberia | SSA | Leprosy | quantitative |
| (Kouassi et al. 2018) | Guinea | SSA | Lymphatic Filariasis | mixed |
| (De Kruijff 2015) | Mozambique | SSA | Leprosy | quantitative |
| (Kumar et al. 2015) | India | Asia | Leprosy | quantitative |
| (Lira et al. 2012) | Brazil | South America | Leprosy | quantitative |
| (Renita et al. 2010) | Brazil | South America | Leprosy | quantitative |
| (Lusli et al. 2015) | India | Asia | Leprosy | mixed |
| (Marahatta et al. 2018) | Nepal | Asia | Leprosy | qualitative |
| (Marega et al. 2019) | Mozambiqu | SSA | Leprosy | quantitative |
| (Marks et al. 2017) | Ghana | SSA | Yaws | mixed |
| (Mues et al. 2014) | India | Asia | Lymphoedema | quantitative |
| (Nsai et al. 2018) | Cameroon | SSA | Buruli ulcer | quantitative |
| (Nwafor et al. 2019) | Nigeria | SSA | Buruli ulcer | quantitative |
| (Ocaya et al. 2015) | Uganda | SSA | Buruli ulcer | quantitative |
| (Gupta 2015) | India | Asia | Leprosy | report |
| (Pearson 2018) | Uganda | SSA | Buruli ulcer | qualitative |
| (Peeters Grietens et al. 2012) | Cameroon | SSA | Buruli ulcer | mixed |
| (Prochazka et al. 2020) | Liberia | SSA | Buruli ulcer, leprosy | qualitative |
| (Pryce et al. 2018) | Nepal | Asia | Lymphatic Filariasis | quantitative |
| (Rai et al. 2020) | Indonesia | Asia | leprosy | qualitative |
| (Singh et al. 2013) | Nepal | Asia | Leprosy | quantitative |
| (Ramos et al. 2012) | Ethiopia | SSA | leprosy | quantitative |
| (Rogers et al. 2018) | Liberia | SSA | Leprosy | quantitative |
| (Sarkar and Pradhan 2016) | Global | Global | Leprosy | review |
| (Sarkar et al. 2012) | India | Asia | leprosy | review |
| (Da Silva and Paz 2019) | Brazil | South America | Leprosy | qualitative |

| (Stanton et al. 2016) | Ghana | SSA | Lymphatic Filariasis | mixed |
|---|---|---|---|---|
| (Stanton et al. 2015) | Malawi | SSA | Lymphatic Filariasis | quantitative |
| (Subedi and Engelbrektsson 2018) | Nepal | Asia | Leprosy | qualitative |
| (Mani et al. 2015) | India | Asia | Leprosy | quantitative |
| (Susanto et al. 2017) | Indonesia | Asia | Leprosy | qualitative |
| (Tabah et al. 2018) | Cameroon | SSA | Leprosy | quantitative |
| (Tembei et al. 2018) | Cameroon | SSA | Leprosy | quantitative |
| (Muthuvel et al. 2017) | India | Asia | Leprosy | mixed |
| (Velink et al. 2016) | Ghana | SSA | Buruli ulcer | mixed |
| (Lal et al. 2017) | India | Asia | Leprosy | quantitative |
| (Wharton-Smith et al. 2019) | Ethiopia | SSA | Mixed NTDs, including lymphatic filariasis | qualitative |
| (Wijeratne and Østbye 2017) | Sri Lanka | Asia | leprosy | quantitative |
| (Ziperstein et al. 2014) | Togo | SSA | Lymphoedema | qualitative |

Results will be presented based on demand-side factors influencing access to health services, followed by the supply-side factors influencing access to health services, as outlined in the Levesque et al. (2013) conceptual framework. In order to identify and highlight those enablers and barriers to access to effective health services along the patient pathway for people affected by SSSDs, see Table 3.

Include Table 3 here.

**Table 3.** Supply and demand aspects of health seeking across the patient pathway described within included papers.

| Study | Ability to Perceive | Health Beliefs | Approachability | Ability to Seek | Acceptability | Ability to Reach | Availability and Accommodation | Ability to Pay/ Affordability | Ability to Engage | Appropriateness |
|---|---|---|---|---|---|---|---|---|---|---|
| (Abdulmalik et al. 2018) | x | X | | | | | | x | x | x |
| (Abedi et al. 2013) | x | | | | | | | x | | |
| (Ackumey et al. 2011) | x | X | x | x | x | x | | | x | X |
| (Adhikari et al. 2015) | x | X | x | x | x | x | | x | x | |
| (Alferink et al. 2013) | x | X | | | | | | | | |
| (John et al. 2010) | x | | x | x | | x | | x | X | |
| (Atre et al. 2011) | x | X | x | | x | | | | | |
| (Dean et al. 2019) | x | X | | x | x | | | x | | X |
| (Dupnik et al. 2013) | | | | | | | | | X | |
| (Garchitorena et al. 2015) | | | | | | | | | | |
| (Gautham et al. 2011) | x | | x | | | x | | x | x | X |
| (Girma et al. 2018) | | | | | | | | | | |
| (Gómez et al. 2018) | x | | | | | | | x | | X |
| (Henry et al. 2016) | x | X | x | x | | | | | | X |
| (Heukelbach et al. 2011) | | | | | | X | | | x | x |
| (Ib et al. 2015) | x | | x | | | | | x | | |
| (Kasang et al. 2019) | | | x | | | x | | | | X |
| (Kouassi et al. 2018) | x | X | x | x | x | | | | | X |
| (De Kruijff 2015) | x | | x | | | x | | | | X |
| (Kumar et al. 2015) | | | x | | | | | | x | X |
| (Lira et al. 2012) | x | | | | x | | | | X | |
| (Renita et al. 2010) | | | | x | | | x | x | x | X |
| (Lusli et al. 2015) | | | x | x | | | | | X | |
| (Marahatta et al. 2018) | | x | | x | x | | | | | X |
| (Marega et al. 2019) | x | X | x | x | x | | | | | X |

| | | | | | | | | | | |
|---|---|---|---|---|---|---|---|---|---|---|
| (Marks et al. 2017) | | x | | | x | | | | x | X | |
| (Mues et al. 2014) | x | | | | | | | | x | x | |
| (Nsai et al. 2018) | | | x | | | | | | | | X |
| (Nwafor et al. 2019) | x | X | | X | | | | | | | |
| (Ocaya et al. 2015) | x | | | x | x | x | | | X | | |
| (Gupta 2015) | | | | | | | | | X | | |
| (Pearson 2018) | x | X | x | x | | | | | | | X |
| (Peeters Grietens et al. 2012) | x | X | x | | x | x | x | | x | | X |
| (Prochazka et al. 2020) | x | | x | x | x | | x | | | x | X |
| (Pryce et al. 2018) | x | | x | x | x | | | | x | | |
| (Rai et al. 2020) | | x | | x | | | | | X | | |
| (Singh et al. 2013) | x | X | x | X | | | | | | | |
| (Ramos et al. 2012) | | | | | | | | | x | | |
| (Rogers et al. 2018) | | | x | | | | | | x | x | x |
| (Sarkar and Pradhan 2016) | | | | x | | | | | x | | |
| (Sarkar et al. 2012) | | | | | | | | | | | |
| (Da Silva and Paz 2019) | | | | | | | | | | | X |
| (Stanton et al. 2016) | x | X | x | x | x | | | | x | x | X |
| (Stanton et al. 2015) | | | x | | | x | | | X | | |
| (Subedi and Engelbrektsson 2018) | x | X | | | | x | | | | | X |
| (Mani et al. 2015) | x | | x | x | | x | | | x | X | |
| (Susanto et al. 2017) | x | X | | x | | | | | X | | |
| (Tabah et al. 2018) | x | X | | X | | | | | | | |
| (Tembei et al. 2018) | | | | | | | | | X | | |
| (Muthuvel et al. 2017) | x | X | x | x | x | x | x | | x | | |
| (Velink et al. 2016) | x | | x | x | x | x | | | | X | |
| (Lal et al. 2017) | x | | x | | | | | | | X | x |
| (Wharton-Smith et al. 2019) | x | x | x | x | X | | | | | | |
| (Wijeratne and Østbye 2017) | | | | | | | | | | | X |
| (Ziperstein et al. 2014) | x | | x | | | x | | | x | x | x |

*4.2. Demand-Side Factors*

The role of supernatural beliefs and stigma emerged strongly as critical aspects influencing demand for health-seeking for patients with SSSDs. Papers also highlighted the role of knowledge as it relates to awareness and care-seeking; and patient confidence in and expectations of the health system. Findings are summarised in accordance with the five demand-side access dimensions.

4.2.1. Ability to Perceive

The ability to perceive focuses on factors that shape an individual's ability to understand that they need to seek health care related to factors such as health literacy and beliefs (Levesque et al. 2013). Thirty-four studies included findings relating to patients' ability to perceive their need to seek care (see Table 3 for papers).

Low knowledge/awareness of diseases and limited perception of severity led to delays in health-seeking in 13 studies (Ackumey et al. 2011; Alferink et al. 2013; Ukwaja et al. 2020; Peeters Grietens et al. 2012; Subedi and Engelbrektsson 2018; John et al. 2010; Dean et al. 2019; Henry et al. 2016; Muthuvel et al. 2017; Ziperstein et al. 2014; Gautham et al. 2011; Susanto et al. 2017). Holding correct health knowledge about an illness, including perception of severity, was found to play an important role in care-seeking. Local communities may have knowledge about SSSDs, e.g., local awareness that Buruli Ulcer (BU) is found along rivers/muddy places (Pearson 2018), but may be unfamiliar with biomedical terms used by health workers (Ocaya et al. 2015; Wharton-Smith et al. 2019; Velink et al. 2016), emphasising the need for community awareness and engagement to ensure familiar local terms are used. Eight studies highlighted that illness was tolerated when perceived as trivial, with patients seeking care when symptoms become severe and/or impact everyday activities (John et al. 2010; Ackumey et al. 2011; Dean et al. 2019; Gómez et al. 2018; Henry et al. 2016; Peeters Grietens et al. 2012; Muthuvel et al. 2017; Ziperstein et al. 2014). For example, in Ghana most respondents with pre-ulcer sought care from the herbalist, while patients with ulcers were more likely to use government facilities and private health practitioner for pain relief (Ackumey et al. 2011). This has implications for preventing permanent impairment associated with leprosy (John et al. 2010; Gautham et al. 2011).

Limited understanding of the need for long-term management and the importance of completion of longer treatment regimens was described across four studies as contributing to limiting long-term management of SSSDs (Lira et al. 2012; Ackumey et al. 2011; Pryce et al. 2018; Kouassi et al. 2018).

4.2.2. Health Beliefs and Their Role in Health-Seeking

Supernatural beliefs about the origin and causation of SSSDs are widespread, and were described in 22 studies (see Table 3). Supernatural reasons described include 'spiritually inflicted illness' (Abdulmalik et al. 2018; Stanton et al. 2015), 'spell' (Tabah et al. 2018), 'evil spirits' (Ackumey et al. 2011), 'witchcraft' (Dean et al. 2019; Henry et al. 2016; Nwafor et al. 2019; Kouassi et al. 2018), 'consequence of punishment for bad deeds' (Marahatta et al. 2018; Singh et al. 2019; Tabah et al. 2018), 'God's desire' (Marega et al. 2019), 'sorcery' (Peeters Grietens et al. 2012; Pearson 2018) and 'superstition' (Rai et al. 2020). In some instances, less pervasive beliefs about disease causation were described, with herbalists themselves emphasising that BU could not be inflicted upon another person (Pearson 2018), and differing beliefs according to patient demographic, with a belief in *'modern medicine'* more common among women and younger participants in a study in Liberia (Dean et al. 2019).

Dealing with the underlying spiritual causation may be considered vital, even if biomedical care is also sought: Dual causality of disease was described in 11 studies with patients believing either simultaneously or in sequence in both biomedical and supernatural causes (Dean et al. 2019; Peeters Grietens et al. 2012; Ackumey et al. 2011; Pearson 2018; Rai et al. 2020; Henry et al. 2016; Adhikari et al. 2015; Ocaya et al. 2015; Marega et al.

2019; Kouassi et al. 2018; Kumar et al. 2015). Four studies (Ackumey et al. 2011; Pearson 2018; Dean et al. 2019; Peeters Grietens et al. 2012) described an oscillation between formal and traditional health systems.

Some supernatural beliefs about the origin of SSSDs place unfair blame on the patient, contributing to stigma and social isolation. It is important that these beliefs are acknowledged and challenged where appropriate to break the cycle of stigma associated with SSSDs. Where patients believe that their illness is their fault or a result of God's desire, there were descriptions of delayed care-seeking (Marega et al. 2019; Marks et al. 2017; Muthuvel et al. 2017).

Patient trust and expectation in the health system is influenced by general awareness of services available, past experiences with treatment-seeking within the health system and local historical context. These must all be taken into consideration with tailored, appropriate measures taken to address underlying reasons for a loss of trust in the health system, where this exists, as highlighted in 11 studies (John et al. 2010; Nwafor et al. 2019; Peeters Grietens et al. 2012; Pryce et al. 2018; Singh et al. 2013; Susanto et al. 2017; Ziperstein et al. 2014; Stanton et al. 2015; Tabah et al. 2018; Adhikari et al. 2015; Atre et al. 2011).

### 4.2.3. Ability to Seek

Ability to seek care relates with having the autonomy to choose to seek care and the knowledge about the options of care available (Levesque et al. 2013). The ability to seek care was described in 25 studies (see Table 3 for papers). Engaging with a range of actors within the community is of importance, particularly in the context of patriarchal household decision-making around care-seeking in some settings. Six papers describe factors influencing personal autonomy for decision-making related to seeking health services (Ackumey et al. 2011; John et al. 2010; Dean et al. 2019; Renita et al. 2010; Pearson 2018; Mani et al. 2015; Muthuvel et al. 2017; Wharton-Smith et al. 2019). These factors include gender, with women needing to seek permission from a husband/male guardian prior to seeking care (John et al. 2010; Dean et al. 2019; Wharton-Smith et al. 2019); preference to discuss care-seeking and/or treatment with family and friends, with parents commonly found to influence care-seeking among young adults (Ackumey et al. 2011; Dean et al. 2019; Pearson 2018).

Stigma emerged as a strong theme within the household and community values. Stigma, and fear of stigma, can have implications for patient care-seeking for an NTD and other illnesses (especially for women), with eight studies describing patients delaying seeking care and/or not disclosing their diagnosis as a consequence of fear of stigma (Sarkar and Pradhan 2016; Marahatta et al. 2018; John et al. 2010; Rai et al. 2020; Velink et al. 2016; Muthuvel et al. 2017; Wharton-Smith et al. 2019). This was described as particularly pervasive for women in three studies, limiting marriage prospects, leading to divorce and/or resulting in an inability to fulfil gendered social roles in India, Nepal and Ethiopia (Muthuvel et al. 2017; Marahatta et al. 2018; Wharton-Smith et al. 2019). Other social factors such as ethnicity, caste and socio-economic status were also described as contributing to the production of social stigma for people affected by leprosy, highlighting how a '*multitude of social and contextual factors together with the disease itself are responsible for a social construct of stigma*' (Marahatta et al. 2018). Stigma or discrimination may create a barrier to some SSSD patients being able to engage with services for other unrelated illnesses. One study in India found that 17.0% of patients with leprosy experienced difficulty in obtaining treatment for other general ailments (Gautham et al. 2011).

### 4.2.4. Ability to Reach

Ability to reach health care relates to mobility, the availability of suitable transportation and flexibility of occupation to enable someone to physically reach a health provider (Levesque et al. 2013). The ability to reach health services was described in 15 studies (see Table 3). Geographic distance and transportation costs can form considerable barriers for patients, due to financial barriers, or challenges created by physical aspects of their condition, particularly when needing to return for repeated visits for management of SSSDs

(Gautham et al. 2011; Peeters Grietens et al. 2012; Stanton et al. 2015; Subedi and Engelbrektsson 2018; John et al. 2010; Ziperstein et al. 2014; Velink et al. 2016; Adhikari et al. 2015; Ackumey et al. 2011; Heukelbach et al. 2011; Kasang et al. 2019; De Kruijff 2015). Indirect costs include transportation costs, feeding costs, loss of earnings, barriers to reaching specialised care, e.g., for BU hospital treatment far from the patient's community (Peeters Grietens et al. 2012; Ackumey et al. 2011).

Where distance is a barrier, patients in two studies described finding alternative solutions closer to home, including seeking care from a traditional healer (Peeters Grietens et al. 2012), and from a private facility (Subedi and Engelbrektsson 2018). This may have implications for early case detection and quality of care received.

Women often experience additional barriers to reach the health facility as a consequence of their domestic duties and care-giving role, which hinders their ability to reach and regularly attend care, if this is not available close to their home (Ackumey et al. 2011; John et al. 2010). Women's work was felt to place them at higher risk of developing deformities in leprosy, due to women being primarily responsible for cooking and other household activities, which may lead to repeated trauma, ulceration and deformities (Sarkar and Pradhan 2016).

### 4.2.5. Ability to Pay

Ability to pay relates to the capacity of a person affected to generate the funds to pay for health services, without catastrophic expenditure (Levesque et al. 2013). Ability to pay was described in 21 studies (see Table 3 for papers). The financial impact to seek care may only be considered a necessity once disease severity increases (Muthuvel et al. 2017), with implications for disease progression (Rai et al. 2020). '*the lack of finances limited their access to health care, thereby worsening their condition and its manifestations*' page 10 (Rai et al. 2020). Financial support from family and community was described as important for enabling individuals to overcome financial barriers (Abdulmalik et al. 2018; Ocaya et al. 2015).

For patients with SSSDs, the ability to pay may be further complicated by the economic implications of having a stigamitising illness, with leprosy-affected persons not being given a job if they have a physical deformity or distancing themselves from others (Da Silva and Paz 2019; Velink et al. 2016). More details regarding costs are described under the affordability section.

### 4.2.6. Ability to Engage

Ability to engage relates to the client's involvement within treatment decisions, along with motivation and commitment to its completion (Levesque et al. 2013). The ability to engage with services was described in 24 studies (see Table 3 for papers). Family and community support were described as playing a vital role in a patient's ability to engage with services and complete treatment/ongoing care (Susanto et al. 2017; Ocaya et al. 2015; Mani et al. 2015; Abdulmalik et al. 2018; Lusli et al. 2015). Support was described as taking several forms, including encouraging support from family (Abdulmalik et al. 2018; Lusli et al. 2015). Community support was in the form of encouragement, prayers and financial aid, which was especially described as having been provided by religious groups (Abdulmalik et al. 2018). Where family support was not present or was misguided, patients at times were lost to follow up; 11.3% of patients with leprosy who were lost to follow up explained that they had no one to accompany them to the hospital (Mani et al. 2015). Given the role that autonomy in both 'ability to seek' care and 'ability to engage', alongside other actors, such as family and community, can play in both these dimensions, there may be some degree of interchangeability between findings for these dimensions.

Factors which were found to contribute to patients' adherence with treatment include medication side-effects and intolerance; medicine shortages and stock outs, advice from family members, financial challenges to accessing services, alcohol consumption and distance to the treatment facility (Dupnik et al. 2013; Gautham et al. 2011; Heukelbach et al. 2011; Sarkar and Pradhan 2016; Mani et al. 2015; Susanto et al. 2017; Ackumey et al. 2011).

Longer-term training or rehabilitation services to ensure that affected persons can complete long-term management of their conditions and to support pain management was described in just four studies (Gupta 2015; Mues et al. 2014; Prochazka et al. 2020; Ziperstein et al. 2014). Positive communication and language in delivery of this rehabilitation support (Gupta 2015); wound cleaning for pain reduction amongst Buruli Ulcer patients (Mues et al. 2014); ensuring understanding of the mechanisms which cause pain or acute attacks among people with lymphoedema (Ziperstein et al. 2014), were described as supporting understanding and adherence.

### 4.3. Supply-Side Factors

Studies also included an emphasis on the five main access dimensions relating to the provision/supply of health services for patients with SSSDs (Approachability; Acceptability; Availability and accommodation; Affordability; Appropriateness) (Levesque et al. 2013).

#### 4.3.1. Approachability

Approachability relates to the perception of affected persons that they can identify, reach and benefit from the services available (Levesque et al. 2013). The approachability of health services was described in 28 studies (see Table 3 for papers). Community-level patient identification was described across multiple studies; who is involved, and how to do this identification varied, with a clear evidence gap in long-term best practices. A range of methods for raising awareness and sharing information were described, including: the role of networks of community educators involving former patients, community representatives, teachers, parents and community members to promote awareness and to advocate for early medical treatment, given the importance of discussions about their condition and whether to seek care with family, friends and other community members among people affected (Ackumey et al. 2011; Tabah et al. 2018); training traditional healers about identification and referral (Adhikari et al. 2015); sharing information through CHWs and/or other health workers (Ackumey et al. 2011; Kasang et al. 2019; Nsai et al. 2018; Prochazka et al. 2020; Pryce et al. 2018; Singh et al. 2013; Rogers et al. 2018; Mani et al. 2015; Velink et al. 2016; Lal et al. 2017; Wharton-Smith et al. 2019; Kouassi et al. 2018; Tabah et al. 2018); active case finding by Community Health Worker (CHW) teams (which increased % of patients identified via this method from 22.0% to 87.0%, and reduced grade II disability at presentation from 13.2% to 7.0%) (Kasang et al. 2019); TV/radio media to share information (Atre et al. 2011; Pryce et al. 2018; Stanton et al. 2015; Singh et al. 2019); self-help groups to identify new symptomatic patients based on their personal experiences with leprosy (De Kruijff 2015); posters and other Information Education Communication (IEC) materials and folk dramas (Atre et al. 2011); retention of knowledge from a previous research study (Stanton et al. 2016). Policy makers should consider the need to continue awareness-raising through diverse and trusted context-specific channels, as locations transition from being endemic, and active case finding ends (Muthuvel et al. 2017).

#### 4.3.2. Acceptability

Acceptability relates to cultural and social factors which influence the acceptability of health services to the person affected (Levesque et al. 2013). The acceptability of health services was described in 17 studies (see Table 3 for papers). Much of the discussion relating to acceptability related to care-seeking from both biomedical and traditional providers, with patients often alternating between types of provider, related to belief in the cause of the illness, type of condition, symptom experienced, proximity of provider, cost of seeking care and previous experiences of care (see Section 4.2.2 health beliefs for more information). This may cause delay in reaching diagnosis and starting treatment, which may have implications for permanent disability as a result. Research in Liberia revealed that the acceptability and choice of both biomedical and traditional providers related to

previous care experiences, patient doubt in the diagnosis provided, and patient refusal to accept that there is no cure for their illness (Dean et al. 2019).

Delayed diagnosis occurred due to multiple encounters within the formal health system before reaching a specialised hospital (Peeters Grietens et al. 2012), and patient seeking care from a private and/or traditional healer prior to the public health system (Srinivas et al. 2018; Atre et al. 2011; Muthuvel et al. 2017). Patient delay was found to be a risk factor for grade 2 leprosy (Srinivas et al. 2018; Muthuvel et al. 2017).

The need for service provision by a health worker of the same gender as the patient, particularly for illnesses which affect genitalia, was highlighted in one study in Ethiopia (Wharton-Smith et al. 2019). "*Women's perceived shame and fear of receiving treatment from a male clinician may also deter women from seeking services from health facilities where the doctors are predominantly male*" page 12 (Wharton-Smith et al. 2019).

Patient understanding of the purpose and duration of treatment, and acknowledgement of the role of medicine and dressings for cure, including potential side-effects, is an important part of acceptance of painful treatments (Lira et al. 2012; Velink et al. 2016; Prochazka et al. 2020).

### 4.3.3. Availability and Accommodation

Availability and accommodation relate to health services being physically available and with enough capacity to provide care (Levesque et al. 2013). There was limited discussion about availability and accommodation, with only four studies addressing it (see Table 3 for papers). Compassion among health workers is critical, with patients choosing to seek less technically suitable care in preference to seeking care from a health worker viewed as unkind. Patient mistrust of the public health system, including the perception that staff lacked compassion and there were long waiting times, acted as deterrents, contributing to patients seeking care from private providers in the first instance (Muthuvel et al. 2017; Srinivas et al. 2018).

One study described NGO provision of additional support which promotes acceptance of services in Cameroon. Additional support includes free inpatient treatment, plus supplementary aid of free meals, free accommodation for inpatients and their caregivers, supplementary schooling and free provision of basic materials for everyday needs—soap, bandages and sheets (Peeters Grietens et al. 2012).

Despite the availability of inpatient treatment services for BU and leprosy, Prochazka et al. highlighted that BU patients feared developing leprosy themselves, preferring to use outpatient rather than inpatient services as a result (Prochazka et al. 2020).

### 4.4.4. Affordability

Affordability relates to the ability of people to be able to use appropriate services; it includes both the direct price of services and opportunity costs relating to loss of income while seeking care (Levesque et al. 2013). Direct and indirect treatment costs were frequently described as creating barriers to seeking care for SSSDs for both formal and traditional treatment. Twenty-one papers presented findings in relation to patients' ability to pay for or the affordability of services (see Table 3 for papers) and there is some interchangeability of findings between these dimensions. Direct costs are perceived to be a barrier to initial health-seeking as a result of minimal information or awareness about the provision of free services (Abdulmalik et al. 2018; Alferink et al. 2013; Peeters Grietens et al. 2012). High direct costs are particularly problematic for expensive treatments such as hydrocele surgery (Adhikari et al. 2015; Stanton et al. 2015). Indirect costs included time away from domestic responsibilities (John et al. 2010; Muthuvel et al. 2017); loss of wages (Gautham et al. 2011; Mani et al. 2015; Muthuvel et al. 2017); inability to pay transport costs to reach services (Gómez et al. 2018; Mani et al. 2015; Muthuvel et al. 2017; Ziperstein et al. 2014).

### 4.4.5. Appropriateness

Appropriateness of care describes the fit between services available and the needs of clients, including what services are provided and the quality of those services (Levesque et al. 2013). The appropriateness of health services was described by 26 studies (see Table 3 for papers). The main factors described with regards to appropriateness include: diagnosis, communication, perception of quality, management, holistic care and referral.

Diagnosis: The ability of the health worker to successfully diagnose a patient's illness is the first step in the pathway towards care for that patient. Eleven studies highlighted deficits in health workers ability to be able to diagnose and manage SSSDs (Henry et al. 2016; Kouassi et al. 2018; De Kruijff 2015; Marega et al. 2019; Rogers et al. 2018; Renita et al. 2010; Nsai et al. 2018; Stanton et al. 2016; Subedi and Engelbrektsson 2018; Wijeratne and Østbye 2017; Ziperstein et al. 2014; Da Silva and Paz 2019), with a clear need for ongoing refresher training of health workers regarding diagnosis and management, to consolidate learning, including knowledge about atypical presentation (Henry et al. 2016; Wijeratne and Østbye 2017). Hierarchical structures created diagnostic barriers in one study, where nurses trained to diagnose leprosy did not have authority to make the diagnosis (De Kruijff 2015). Differences in diagnostic pathways within the same country (Liberia) highlighted a lack of standardisation of care (Kasang et al. 2019; Prochazka et al. 2020). As the number of cases of a particular SSSD reduces, the health workers' ability to correctly diagnose and manage patients may also dwindle. Additionally, if health workers move from non-endemic to endemic areas within a country, knowledge and skills may be lacking (De Kruijff 2015; Nsai et al. 2018).

Communication: Health workers play a critical role in communicating and ensuring that their patients understand the realities of their condition (whether reversible or permanent), future anticipated ongoing treatment and care needs. Poor communication between the health worker and patients creates a barrier to treatment completion, as described by a patient with Buruli Ulcer in Cameroon. "*I spent months there at the hospital, but I could only endure so much. They treat you like a dog. But I'm a man. So finally I had no choice but to leave*" page 6 (Peeters Grietens et al. 2012).

Health workers should be clear about anticipated treatment durations, so that patients have clear expectations for the pathway towards recovery to avoid misunderstanding (Dean et al. 2019; Peeters Grietens et al. 2012; Gautham et al. 2011; Atre et al. 2011).

Patient perception of quality: One theme that emerged was patient perception of quality, which included the importance of timeliness of recovery and slowing of disease progression for a treatment to be considered effective (Ackumey et al. 2011; Prochazka et al. 2020). "*Treatment was considered effective when it fulfilled respondents' expectations of slowing disease progression and recovery*". (Ackumey et al. 2011).

Management: Supply-chain planning should ensure adequate available stock to build patient confidence to regularly attend treatment. Lack of needed drugs and supplies has implications for patients starting treatment (Dean et al. 2019), as longer duration before starting treatment can have implications for disease progression (Heukelbach et al. 2011) and may lead to patients seeking care from alternative providers, such as herbalists, rather than at a health facility (Pearson 2018). Supply-chain management should include stock needed to manage reactions and side-effects, in addition to the treatment itself, for example, ensuring adequate quantities of steroid drug (prednisolone) needed to manage leprosy reaction (Gautham et al. 2011; De Kruijff 2015).

Holistic care: Health workforce planning should take into consideration the range of services needed for holistic patient-centred care, including wound management, physiotherapy and mental health, as well as drug treatment (Stanton et al. 2016, prochazka). Research by Prochazka et al. (2020) highlighted understaffing and need for physical therapy for patients with BU and leprosy at a specialised site in Liberia.

Only three included studies focused on mental health service provision (Abdulmalik et al. 2018; Gautham et al. 2011; Dean et al. 2019), highlighting the need for greater focus on provision of these services for SSSD patients to ensure holistic care, particularly since patients described a decline in their mental health around the time of realising that their

condition was permanent (Dean et al. 2019). "*a few participants who had been told that lymphoedema was irreversible or that there was 'no cure' linked this understanding of the permanency of their condition to deterioration in their mental health*". (Dean et al. 2019).

Referral: Referral services are often limited, with a reported lack of good coordination of care for patients with SSSDs (Kruijff, Stanton). Eye care, surgical care, and psychosocial support were the main referral reasons described (De Kruijff 2015; Prochazka et al. 2020). General health systems strengthening is needed to ensure that specialist services are available when needed. Even where referral channels officially exist, e.g., for surgical management including debridement and in some cases amputation at a neighbouring hospital in Liberia, these referrals were discouraged by staff at the referral hospital, due to their fear of BU patients. "…*staff complained that they were discouraged from organising referrals because of resistance from staff at receiving facilities who feared disease transmission or cited limited resources to cover surgical costs*". (Prochazka et al. 2020).

Even where referral is made, patients may not take up the referral, due to lack of transportation (Ziperstein et al. 2014), lack of confidence in the effectiveness of biomedical care, due to a perceived lack of effectiveness of biomedical care as compared with care provided by an herbalist (Pearson 2018).

## 5. Discussion

This scoping review has revealed a wealth of information surrounding the various enablers and barriers which can exist across the patient pathway, from both supply and demand aspects of health services for people affected by SSSDs, along with the interlinkages across and between these, according to the aspects of the Levesque et al. (2013) conceptual framework. The importance of spiritual beliefs in determining care-seeking and as an underlying factor in stigma and discrimination has emerged strongly as a key finding from this review. The main focus of the majority of included papers was on the curative aspect of care for affected patients. There were few papers which included emphasis on holistic care, including physical rehabilitation and psychosocial support. Similar barriers to seeking health care were described across disease conditions. However, many of these studies have, to date, focused on leprosy. There is a need for future research which focuses on multiple SSSDs.

Early diagnosis and care-seeking should be prioritised to reduce permanent morbidity and consequential disability. Creating strong knowledge and understanding about SSSDs, along with the importance of early care-seeking to prevent permanent impairment, is critical. This review demonstrated evidence that involving CHWs in community case-finding can strengthen links for referral, reducing delays in care-seeking decisions and increasing patient enrolment, with reduced permanent impairment at diagnosis (Kasang et al. 2019). CHWs can also play a role in addressing geographic barriers to reaching care, through provision of home-based services (Kasang et al. 2019). This is in keeping with a range of other papers which outline the key roles CHWs, community drug distributors and other trust actors in the community play in the provision of NTD-related services (Krentel et al. 2017; Vouking et al. 2013; Oluwole et al. 2019). However, placing additional burden on these cadres in the provision of health services needs to be carefully balanced with a strengthening of systems to support CHWs to effectively deliver their critical function (Kok et al. 2014).

Given the widely described importance of spiritual beliefs within the care-seeking pathway for patients affected by SSSDs, there is value in endeavouring to understand these factors and consider opportunities to engage with informal health providers to support identification and early referral of patients affected by SSSDs for biomedical care (Adhikari et al. 2015). Despite traditional healers being the first point of contact for many patients with SSSDs, only one study described them as being a source of information and awareness (Adhikari et al. 2015). Working with traditional healers could be an untapped opportunity to share key information and awareness about SSSDs within communities and is a key avenue that will be explored further within the REDRESS programme.

Addressing underlying community stigma and discrimination is crucial, since fear of social isolation following a diagnosis can lead to delays in seeking care, and opportunities for integrating this within SSSD services should be prioritised, in keeping with recommendations within the WHO NTD Roadmap (WHO 2020).

There is a need to engage with communities to promote understanding about their rights regarding the range of services available, including routine services available, as well as campaign activities (Stanton et al. 2015). Awareness-raising should highlight the range of services available to patients, including treatment, counselling, self-help groups, and home-based care services, where these are available. Information provided to patients should include clear details about potential complications, and what to do to reduce patients being lost to follow up (Mani et al. 2015; Lal et al. 2017).

Generating demand for health care-seeking must be accompanied by strong health systems, which have well-trained, supervised and respectful staff available to correctly diagnose SSSDs, and who are equipped with the needed drugs and supplies to consistently manage patients throughout treatment for the SSSD, and beyond, depending upon patient needs (REDRESS unpublished HR literature review).

The absence of evidence surrounding the reasons for supply-chain gaps relating to SSSD care (Dean et al. 2019) highlights the need for research to study and explore supply-chain challenges, in order to help provide tailored responses to bottlenecks. The evidence that does exist relates to the supply chain for Mass Drug Administration programmes (Villacorta Linaza et al. 2021), rather than the ongoing supply-chain needs associated with case management programmes. Key recommendations include reinforcing the role of qualified pharmacists within NTD programmes, along with ensuring the integration of the NTD supply chain within national health systems (Villacorta Linaza et al. 2021; WHO 2020). If the supply chain is not consistent, then demand generation efforts will be undermined by patient loss of confidence and trust in the health system, which may reinforce patient care-seeking from alternative sources, such as herbalists and traditional healers.

This scoping review has identified that much of the literature regarding health-seeking for patients affected by SSSDs has focused on curative treatment, such as drugs and dressings. We found limited evidence surrounding holistic care, including appropriate referral for other services, such as eye care and surgical care, physical rehabilitation services and psychosocial support, with mental health care where needed. Specialists, who may irregularly care for patients with SSSDs, may need training about SSSD care in order to avoid discriminatory behaviour. Notable by its near absence was the literature surrounding ongoing disability support for patients with a disability relating to NTD. Since millions of people are affected by disabilities as a consequence of NTDs (Kuper 2021), such as visual and physical impairments (Kuper 2019), this is a critical research gap and priority area for future study, in order to seek to ensure the inclusion of people with a disability within NTD programmes (Kuper 2019). Care for people affected should include rehabilitation which includes both clinical services (such as physiotherapy and assistive devices) and also actions to improve employment, overcome stigma and enhance social participation of people with disabilities (Kuper 2019). The inclusion of people with a disability in developing holistic solutions for NTD programmes and health systems alike should be prioritised.

The need for mental health screening and psychosocial support is pertinent, particularly in light of WHO's recognition that NTDs are major drivers of mental ill-health (World Health Organization 2020). Furthermore, a recent systematic review found that among people affected by leprosy, LF and podoconiosis, the burden of mental illness ranged from 12.6% up to 71.7%, with high levels of suicidal ideation (18.5%) (Ali et al. 2021). There is a need to consider and provide holistic services, which include psychosocial support and an emphasis on patient wellbeing (World Health Organization 2020). Furthermore, in light of the changing funding environment, such as the reduction in UKAID, which creates uncertainty surrounding the sustainability of NTD service provision and threatens progress already made (Sightsavers 2021), there is more need than ever

for comprehensive understanding of the barriers and enablers to patient access to health services for SSSDs, to ensure that activities and interventions make the most efficient use of this reduced funding, in order to best ensure services reach those who are most marginalised.

## 6. Conclusions

This review has highlighted the aspects of the patient-care pathway in line with the Levesque framework for access to health care, particularly important to understand for marginalised populations, including many people affected by SSSDs. The Levesque pathway formed a valuable and systematic framework within which to consider the various factors which influence health-seeking for people affected by SSSDs. The role of supernatural beliefs in patient knowledge and understanding and in care-seeking practices emerged as a critical feature of health-seeking for patients with SSSDs. This interconnects with stigma, another critical feature which has influence on care-seeking, mental health and wellbeing for the patient and their family. Strengthening the patient-care pathway will require greater engagement with community actors to initially identify and refer patients with SSSDs, and then to repeatedly encounter and encourage ongoing continuation of treatment where necessary, with adequate consideration of rehabilitative and mental health needs.

**Author Contributions:** Conceptualization, R.M., H.B., and L.D.; methodology, R.M., H.B., and L.D.; software, R.M., H.B., and L.D.; validation, R.M., H.B., and L.D.; formal analysis, R.M., H.B., and L.D.; writing—original draft preparation, R.M.; writing—review and editing, R.M., H.B., S.T., Z.Z., K.K. and L.D.; funding acquisition, L.D., K.K., Z.Z., and S.T. All authors have read and agreed to the published version of the manuscript.

**Funding:** This work was supported by the National Institute for Health Research (NIHR), Research for Innovation for Global Health Transformation REDRESS Programme (NIHR2001129).

**Institutional Review Board Statement:** The study was approved by both the Liverpool School of Tropical Medicine ethics review board (protocol ID 20-040) and the University of Liberia Institutional Review Committee (UL-PIRE IRB) (protocol ID 20-09-233).

**Informed Consent Statement:** Not applicable.

**Data Availability Statement:** Not applicable.

**Conflicts of Interest:** The authors declare no conflict of interest. The funders had no role in the design of the study; in the collection, analyses, or interpretation of data; in the writing of the manuscript, or in the decision to publish the results.

## Appendix A

Details the search terms used for this literature review.

**Review Search terms**

We used the following key words and MeSH terms for searching the databases

|   | Topic specific key word | Mesh term | Free text term |
|---|---|---|---|
| OR | Access | Health services accessibility | Health N3 Access |
| OR | Availability | | Health N3 Availability |
| OR | Help | Helping behaviour | Help* |
| | | Help-seeking behaviour | |
| | | Help behaviour | |
| OR | Utilisation | Patient acceptance of health care | Health N3 Utili* |
| | | | Health N3 use |
| OR | Behaviour | | health N3 Behaviour |
| | | | health N3 behavior |

| OR | Seek | | Health N3 Seek* |
|----|------|------|------|
| | | | Health N3 accept* |
| OR | Practice | Health knowledge, attitudes, practice | Health N3 practice |
| OR | Perception | | Health N3 perception |
| OR | Attitude | Attitude to health | Health N3 attitude |
| OR | Belief | | "health belief*" |
| OR | Enabler | | Enabler or motivator or facilitator |
| OR | Barrier | | Barrier |
| OR | Demand | Health service needs and demands | Health N3 need |
| | | | Health N3 demand |

**AND**

| | Topic specific key word | Mesh term | Free Text Term |
|----|------|------|------|
| OR | skin disease | Skin diseases, infectious | |
| | | Skin diseases, parasitic | |
| OR | Neglected Tropical Diseases | Neglected diseases | |
| OR | Leprosy | Leprosy | Leprosy |
| | | Mycobacterium leprae | "Hansen's disease" |
| | | | "Mycobacterium Leprae" |
| OR | Buruli ulcer | Buruli ulcer | "buruli ulcer" |
| | | | "mycobacterium ulcerans" |
| | | | "skin ulcer" |
| OR | Yaws | Yaws | Yaws |
| | | | Frambesia |
| OR | Lymphoedema | Lymphedema | Lymphedema |
| | | | lymphoedema |
| | | | "Milroy's disease" |
| OR | Lymphedema | Lymphoedema | Lymphedema |
| | | | lymphoedema |
| | | | "Milroy's disease" |
| OR | Lymphatic filariasis | Elephantiasis, Filarial | "filarial elephantiasis" |
| | | | "lymphatic filariasis" |
| OR | Hydrocele | Testicular hydrocele | Hydrocele |

**AND**

| | Context specific key word | Mesh term | Free text terms |
|----|------|------|------|
| OR | Sub Saharan Africa | Africa south of the sahara | "sub-saharan Africa" |
| OR | Africa | | Africa |

| | | | |
|---|---|---|---|
| OR | Low-and Middle- Income Countries | | LMIC |
| | | | "low and middle income" |
| | | | "low resource" |
| OR | Liberia | Liberia | Liberia |
| OR | Developing countries | Developing countries | "developing countr*" |

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
