# Peer review of "Barriers and Enablers to Health-Seeking for People Affected by Severe Stigmatising Skin Diseases (SSSDs): A Scoping Review"

_socsci, doi:10.3390/socsci11080332_

Round 1

Reviewer 1 Report

This is an important review and an exceptionally well written paper. I enjoyed reading it. I have only one concern, and a few (minor) suggestions for improvement. My concern is that the search was conducted in May 2020 (lines 76-77), which is two years ago. It would be good to update the search.

Minor comments include:
Line 34 ‘NTDs are often neglected’ – neglected tropical diseases are often neglected, maybe the authors can add what they mean by neglected – by whom, in what way? Alternatively, it can be omitted.

Line 47/overall: I wonder if ‘severe stigmatising skin NTDs’ would be more appropriate than SSSDs. Some stigmatised skin NTDs (such as l
eishmaniasis) have not been included, which makes sense because the study was conducted as part of another study that does not include leishmaniasis for example. However, it would be good to add what diseases have not been included, despite being a stigmatised skin NTD – to give the reader an idea of the scope of the review.

Line 63: It would be good to add the research question (the rationale and aim have indeed been described already).

Lines 68, 97, 101, 107, 109, 144, 161, 210, etc.: table, excel, figure, buruli, god – please add a capital letter (Table, Excel, Figure, Buruli, etc.).

Line 69: Good to state, in the main text, that the ‘individual country names’ have not been added to the search strings as well. Liberia is mentioned separately, which makes sense because the study was conducted as part of another study, but is also a limitation in my opinion. It would be good to add a few lines about this (given that the authors say they focus on LMIC). I wonder if any articles have been missed because of this.

Line 89: “Studies published between 2010 and 2020.” It is unclear why this time period has been chosen. Please clarify this.

Lines 95-96: “Three reviewers (HB, LD, RM) then screened titles and abstracts for relevance” – please add whether all titles/abstracts have been screened by all authors, or whether the studies were ‘divided’ between the authors. In addition, please add whether the full-text screening etc. (lines 99 and 101) have been conducted by all three reviewers or not. This is not clear at the moment.

Line 144: “three studies were for a combination of NTDs”. I assume this can be found in Table 3, however, while Table 3 is mentioned 11 times, I don’t see it in the manuscript or the supporting file…

Line 169: BU – I don’t think the abbreviation has been introduced before. The authors later use ‘BUD’ (line 263). Same for CHWs (line 331) and IEC (line 340). Other general comment: sometimes the authors use one decimal place, sometimes none (e.g. 13.2% and 7%, lines 336-337). Please check line 340: “(Are)” and line 374 “caapcity”.

Line 299: it is unclear what ‘community support’ refers to, perhaps this can be clarified.

Lines 314-316: Please check this sentence (Guptawound, Mateo, ziperstein?)

Line 329: ‘discussing with’ – not very clear, who is discussing what?

Line 380: does this relate to compassion or long waiting times?

Discussion: the discussion is written very well, however, it would be good to include a ‘limitations’ section.

Reviewer 2 Report

This is an important scoping review on the barriers and enablers to health seeking for people affected by  severe stigmatising skin diseases. It identified number of significant aspects of patient care pathways. The finding about the role of supernatural beliefs in patient knowledge and understanding is particularly  significant. Levesque framework was appropriate. However,  there were few interchangeable issues  between the components  like ‘ability to pay’, ’ ability to engage’,  and ‘ability to seek autonomy’, on which the authors could make comments.
